# Computational Chemistry-Guided Syntheses and Crystal Structures of the Heavier Lanthanide Hydride Oxides DyHO, ErHO, and LuHO

**Nicolas Zapp** [1] , **Denis Sheptyakov** [2] **and Holger Kohlmann** [1],*

1 Institute of Inorganic Chemistry, Leipzig University, Johannisallee 29, 04103 Leipzig, Germany; nicolas.zapp@uni-leipzig.de

2 Laboratory for Neutron Scattering and Imaging, Paul Scherrer Institut, 5232 Villigen, Switzerland; denis.cheptiakov@psi.ch

\* Correspondence: holger.kohlmann@uni-leipzig.de; Tel.: +49-(0)341-9736201

**Abstract:** Heteroanionic hydrides offer great possibilities in the design of functional materials. For ternary rare earth hydride oxide *RE*HO, several modifications were reported with indications for a significant phase width with respect to H and O of the cubic representatives. We obtained DyHO and ErHO as well as the thus far elusive LuHO from solid-state reactions of $RE_2O_3$ and $RE$H$_3$ or LuH$_3$ with CaO and investigated their crystal structures by neutron and X-ray powder diffraction. While DyHO, ErHO, and LuHO adopted the cubic anion-ordered half-Heusler LiAlSi structure type ($F\bar{4}3m$, $a$(DyHO) = 5.30945(10) Å, $a$(ErHO) = 5.24615(7) Å, $a$(LuHO) = 5.171591(13) Å), LuHO additionally formed the orthorhombic anti-LiMgN structure type (*Pnma*; LuHO: $a$ = 7.3493(7) Å, $b$ = 3.6747(4) Å, $c$ = 5.1985(3) Å; LuDO: $a$ = 7.3116(16) Å, $b$ = 3.6492(8) Å, $c$ = 5.2021(7) Å). A comparison of the cubic compounds' lattice parameters enabled a significant distinction between *RE*HO and $RE$H$_{1+2x}$O$_{1-x}$ ($x$ < 0 or $x$ > 0). Furthermore, a computational chemistry study revealed the formation of *RE*HO compounds of the smallest rare earth elements to be disfavored in comparison to the sesquioxides, which is why they may only be obtained by mild synthesis conditions.

**Keywords:** metal hydrides; oxides; neutron diffraction; ab initio calculations; solid-state synthesis





## 1. Introduction

The hydride ion H$^-$ has a unique chemical character due to its medium electronegativity, high polarizability, and small size, and is thus a promising candidate in the design of functional materials. As stable ionic hydrides are only known for compounds of lowly electronegative cations, e.g., alkaline and alkaline earth elements, the chemical space of binary ionic hydrides is rather limited. In recent years however, a rising number of so-called heteroanionic or mixed anionic hydrides is being investigated that contain additional anions and thus expand the accessible chemical space of ionic hydrides [1–4]. These compounds show several interesting properties such as hydride ion conductivity [5–11], catalytic activity [12–16], superconductivity [17,18], and luminescence [19–25] or photochromism [26–30].

Ternary rare earth hydride oxides (or oxyhydrides) LaH$_x$O ($x$ = 0.78), CeH$_x$O ($x$ = 0.90), PrH$_x$O ($x$ = 0.56), $RE$H$_{1+2x}$O$_{1-x}$ (*RE*: rare earth element; *RE* = La, Ce, Pr; 0 < $x$ ≤ 0.2), and LaH$_{1+x}$O$_{1-y}$ (0 < $y$ ≤ 0.2, $x$ < 2$y$) were first reported in the 1960s and 1980s [31,32]. In all cases, the CaF$_2$ (fluorite) structure type with space group $Fm\bar{3}m$ was assigned, similar to the cubic rare earth oxide fluorides [33]. The crystal structure can be described as a cubic close packing of rare earth atoms whose tetrahedral interstices are occupied by hydrogen and oxygen in a disordered fashion. Recently, the structure of LaH$_{1+2x}$O$_{1-x}$ and its compositional range with respect to H and O were further investigated, and a tetragonal superstructure was assigned. It shows a broad phase width with increasing occupation

of the octahedral interstice by hydrogen [6,34], similar to $LaH_{2+x}$ [35], and full mixing of anions. The 1:1:1 compounds *RE*HO (sometimes called "stoichiometric" hydride oxides, *RE* = La, Ce, Pr, Nd) form a $CaF_2$ superstructure with tetragonal symmetry and space group *P4/nmm*, which was later confirmed by neutron diffraction analysis on LaHO and NdHO (Figure 1, bottom center) [7,31,36,37]. A recent study furthermore observed a monoclinically distorted modification with space group $P12_1/m1$ [34]. At high pressures, the tetragonal LaHO structure transforms to the hexagonal anti-$Fe_2P$ type via the orthorhombic $PbCl_2$ type structure [34,38]. Upon heating, tetragonal LaHO showed a reduction in the intensities of the fluorite superstructure reflections similar to LaOF [39], which is why a fluorite type high-temperature modification was assumed [31]. For the lanthanum compounds, the composition had a great influence on the transport properties: while LaHO showed low and $LaH_{1+2x}O_{1-x}$ high ionic conductivity depending on *x*, $LaH_{1+x}O_{1-y}$ was metallically conducting [6,31].

The smaller rare earth elements crystallize in the cubic fluorite structure type with disordered anions ($Fm\bar{3}m$, *RE* = Y, Sm, Gd–Er; confirmed by neutron powder diffraction analysis on YHO, HoHO, ErHO) [7,12,20,40,41] or ordered varieties thereof, the so-called half-Heusler LiAlSi structure type with space group $F\bar{4}3m$ (*RE* = Ho) [42] and the orthorhombically distorted anti-LiMgN structure type with space group *Pnma* (*RE* = Y; also observed in monoclinic distortion with space group $P12_1/m1$ similar to the lanthanum compound (see above); Figure 1) [34,41]. The phase width of these compounds was investigated on the yttrium system Y–H–O [34]. It is significantly smaller than the analogous lanthanum system due to the less efficient shielding of anionic charges during occupation of the octahedral interstices. The reported compounds $YH_{1+2x}O_{1-x}$ are isotypic to the lanthanum representatives above. Oxygen-rich phases, e.g., $REH_{1-2x}O_{1+x}$, have not been observed yet [34].

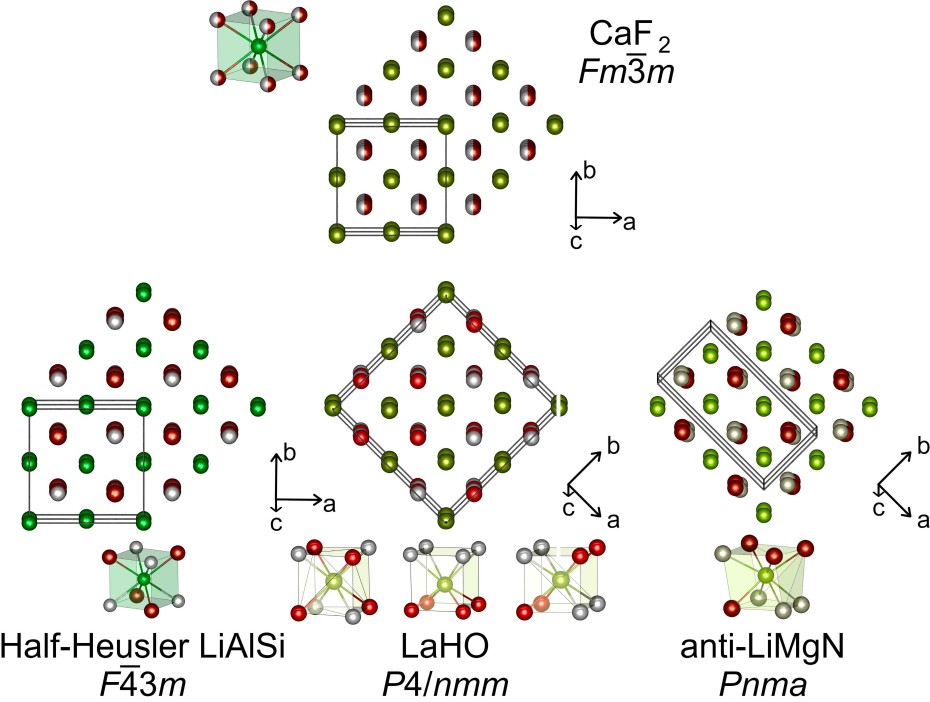

**Figure 1.** The crystal structures of *RE*HO modifications with metal coordination polyhedra, structure types and space group types. The monoclinic LaHO modification with space group $P12_1/m1$ is a slightly distorted variety of the presented anti-LiMgN structure type (*translationengleiche* transition of index 2) [34]. All structure types are crystallographically related [42]. Color code: *RE*, green; H, white; O, red. Reprinted with permission from *Inorg. Chem.* 2021, 60, 6, 3972–3979. Copyright (2021) American Chemical Society.

Though not explicitly studied, several indications were reported for a significant phase width of the anion-disordered fluorite-type $RE$HO compounds: The samarium, dysprosium, and holmium representatives showed two distinct phases that differed in their lattice parameter $a$ in a synchrotron X-ray powder diffraction study (Sm: $\Delta a/a = 0.22\%$, Dy: $\Delta a/a = 0.09\%$, Ho: $\Delta a/a = 0.04\%$) [12]. A greater discrepancy was also reported for two gadolinium compounds, although in different research groups ($\Delta a/a = 0.35\%$) [12,20]. Differences in similar magnitude were observed for the anion-ordered and disordered holmium compounds ($\Delta a/a = 0.65\%$) [12,42], whose anion order was verified by neutron diffraction measurements. The anion-disordered compound additionally showed a significant deviation of a 1:1:1 stoichiometry in the refinement of neutron diffraction data ($SOF$(H) $= 1 - SOF$(O) $= 0.526(2)$) [12] (please note that lifting the restriction of full occupation of crystallographic sites ($SOF$(H) $+ SOF$(O) $= 1$) enables further structural models in the refinement, e.g., an anion-deficient hydride oxide including vacancies). In a similar study of the anion-disordered erbium compound, such a deviation was not observed ($SOF$(H) $= 0.494(3)$, $SOF$(O) $= 0.505(1)$) [7]. Furthermore, several studies verify a broad compositional range of thin film rare earth hydride oxides [26,27].

The identification of such phase widths is an important issue, as it has a major impact on a compound's properties, especially for ion conductivity. From a crystallographic point of view, the formation of hydrogen-rich phases $RE$H$_{1+2x}$O$_{1-x}$ ($RE$X$_{2+x}$; X = H and O) demand the occupation of further crystallographic sites by anions as a full occupation of the tetrahedral interstices of the metal lattice by anions corresponds to a ratio $RE$X$_2$. This is most probably the octahedral interstice, due to its size. In the oxygen-rich counterparts $RE$H$_{1-2x}$O$_{1+x}$ ($RE$X$_{2-x}$), voids must be formed. As the migration of anions in fluorite-related materials happens via octahedral interstices or voids [7,34,43], their quantity directly affects the ionic transport.

The differentiation between anion-ordered and -disordered fluorite phases is of similar importance. A recent literature example observed a difference between conducting and isolating behavior, dependent on the anion ordering in Ba$_2$YHO$_3$ and Ba$_2$ScHO$_3$ [9]. Unfortunately, a reliable differentiation between the anion-ordered and disordered cubic $RE$HO modifications ($Fm\overline{3}m$ vs. $F\overline{4}3m$) is difficult by laboratory X-ray data and best performed with neutron diffraction analysis [42]. In our previous study on HoHO [42], the lattice parameter proved to be easily accessible and observable for differentiation here, which is why we wanted to expand our study to holmium's nearest neighbors in the periodic table, dysprosium, and erbium in order to elaborate a simple tool for a general differentiation of ordered and disordered cubic $RE$HO compounds. Furthermore, the representatives of the smallest rare earth elements Sc, Tm, Yb, and Lu are still missing, though DFT calculations predicted their presence [41,42]. We therefore attempted the synthesis of LuHO by a different synthesis technique starting from LuH$_3$ and CaO, which was previously successful for YHO [41].

Note on Nomenclature: Metal–hydrogen compounds are called metal hydrides. The natural isotopic mixture of hydrogen gas consists of 99.985% $^1$H (protium) + 0.015% $^2$H (deuterium, D). In this manuscript, the term metal hydride is used as a collective name including all isotopes, i.e., protides, deuterides, and tritides. They are only differentiated explicitly when isotope effects are important, for example in neutron diffraction experiments. This nomenclature is used accordingly for metal hydride oxides.

## 2. Materials and Methods

Synthesis. All manipulations were performed in an argon-filled glove box (Air Liquide, Düsseldorf, Germany; 99.999% purity), whose water and oxygen levels were kept below 1 ppm. DyHO, ErHO, and LuHO were obtained from the trihydrides and sesquioxides ($RE$H$_3$ + $RE_2$O$_3$ → 3 $RE$HO) with an excess of hydride compensating its thermal decomposition, and LuHO and LuDO from the reaction of the trihydride (-deuteride) and calcium oxide (LuH$_3$ + CaO → LuHO + CaH$_2$) with a slight CaO excess. The sesquioxides (Kristallhandel Kelpin, Er$_2$O$_3$: 99.9%) were pretreated thermally prior to use at 1425 K for

6 h in a corundum crucible in air and transferred while hot into the glove box [44]. The trihydrides were obtained from the hydrogenation of 2–5 g metal ingots (Dy, Er: Smart Elements, Vienna, Austria, 99.9%; Lu: Onyxmet, Olsztyn, Poland, 99.95%) with hydrogen (Air Liquide, Düsseldorf, Germany 99.9%) or deuterium gas (Air Liquide, Düsseldorf, Germany 99.8%) at 4–8 MPa gas pressure in sealed home-built autoclaves made from Inconel (Böhler L718V). Samples were initially heated to 725 K for 2 h and annealed subsequently for 48 h at 525 K. The thermal decomposition of lanthanide trihydrides under formation of $H_2$ gas requires the choice of a reaction vessel with sufficient material strength to endure the gas pressures and an aligned ratio of trihydride amount to container (gas) volume. For the synthesis of the hydride oxides, silica glass ampoules (10.5 mm inner diameter, 1.2 mm wall thickness) were cleaned with diluted nitric acid and acetone, stored at 450 K for several days, and transferred while hot into the glove box. The starting materials were mixed, ground, and filled into the silica glass ampoules, which were fused subsequently. For DyHO and ErHO, 1.8 g trihydride and 1.5 g oxide (molar ratio 2:1) were sealed in a tube of 20 cm length, heated with 50 K h$^{-1}$ to 1175 K, and annealed for 48 h before resuming to room temperature with 25 K h$^{-1}$. For the reaction $LuH_3 + Lu_2O_3 \rightarrow 3\ LuHO$, 200 mg trihydride and 220 mg oxide (molar ratio 2:1) were sealed in a tube of 7 cm length, heated with 100 K h$^{-1}$ to 1050 K, and resumed to 550 K with 10 K h$^{-1}$ before switching off the furnace. For the reaction $LuH_3 + CaO \rightarrow LuHO + CaH_2$, 700 mg trihydride (2.0 g trideuteride) and 230 mg (700 mg) CaO (molar ratio 1:1.1) were sealed in silica glass tubes of 6 cm (20 cm) length, heated with 25 K h$^{-1}$ to 800 K, and annealed for 48 h (96 h) before switching off the furnace. For the hydride, this procedure was repeated after grinding, and the sample was washed by stirring for 20 min in 60 mL of a $NH_4Cl$ solution in methanol in air, filtrated, rinsed twice with a total of 40 mL methanol, and vacuum dried before being transferring back into the glovebox.

Neutron powder diffraction (NPD) analysis. NPD data were collected on the HRPT high-resolution diffractometer at Paul-Scherrer Institut (PSI, Villigen, Switzerland) with a wavelength of 1.155 Å (DyHO, ErHO) and 1.494 Å (LuDO) [45]. Specimen were filled in vanadium containers of 8 mm diameter in a helium glove box with controlled $H_2O$ and $O_2$ impurities < 1 ppm, sealed with indium gaskets, and measured at ambient temperature. The high absorption of thermal neutrons of the dysprosium sample was addressed by longer counting times.

X-ray powder diffraction (XRPD) analysis. Specimen for XRPD experiments were mixed with powdered diamond to reduce X-ray absorption and filled into Lindemann glass capillaries of 0.2 mm diameter, which were subsequently sealed. They were analyzed by a Stoe STADI-P diffractometer (Stoe & Cie GmbH, Darmstadt, Germany) with Debye-Scherrer geometry and Ge(111) monochromatized Cu-$K_{\alpha1}$ radiation. The data for cubic LuHO were collected in two experiments for small (20–92° 2θ) and large angles (89–128° 2θ).

Rietveld refinement. Powder diffraction data were refined with the Rietveld method [46,47] as implemented in the software GSAS-II version 4673 [48] (NPD data) or Topas version 5 (Bruker AXS, Billerica, MA, USA, XRPD data) employing the fundamental parameter approach. Instrumental functions were provided by the PSI for the former and determined empirically on a Si NIST 640d reference material for the latter. The background was simulated with Chebychev type polynomials of second (LuDO), fourth (NPD of ErHO, LuDO; XRPD data), or fifth (NPD of DyHO) order and single line fits for the contribution of the X-ray capillary material. In each case, the zero-point error, scaling factors, lattice, profile, and positional parameters were refined in addition to the isotropic displacement factor $B_{iso}$, unless the phase ratio deceeded 10 wt%, in which case only the former four parameters were considered. For XRPD data, the $B_{iso}$ values were corrected with an offset value $B_{overall}$, which was refined from the diamond reflections using a literature value for diamond ($B_{iso}(C) = 0.142$ Å$^2$ [49]). The neutron absorption of the dysprosium sample was considered by calculating the absorption parameter $\mu R$ with the FRM-II neutron calculator assuming an effective mass density of half the crystallographic mass density ($\mu R = 4.5$, fixed

during refinement) [50]. It should be noted that GSAS-II regards the energy dependence of erbium's coherent scattering length ($b_c$(Er) = 8.09 fm instead of 7.79 fm at 1.798 Å [51,52]).

Density functional theory (DFT) calculations. The density functional theory calculations were performed with the Vienna ab intio simulation package (VASP) version 5.4.4 [53,54] using projector augmented wave pseudopotentials (PAWs) [55] from the VASP database. The *f* electrons of lanthanides *Ln* were described as part of the frozen core (*Ln*_3). The *s* and *p* level electrons of the second highest principal quantum number were treated as semi-core states of Li, Ca, and Y (Li_sv, Ca_sv, Y_sv). For La, H, O, and F, the standard potentials were employed (La, H, O, F). The GGA-PBE method [56] described the exchange correlation potential, and the Brillouin zone was integrated using the tetrahedron method with Blöchl corrections [57]. The *k*-grid was generated automatically and Γ-centered, ensuring a *k*-point density of 0.03 Å$^{-1}$ in each direction (LiH, LiF, CaO 8 × 8 × 8; CaH$_2$ 6 × 10 × 6; *RE*H$_3$ 6 × 6 × 5; *RE*$_2$O$_3$ 3 × 3 × 3; tetragonal *RE*HO 4 × 4 × 6; orthorhombic *RE*HO 4 × 9 × 6; cubic *RE*HO 6 × 6 × 6; *RE*OF 10 × 10 × 2). The cutoff energy was set to 800 eV. Structure optimizations were carried out in accurate precision mode and with full degrees of freedom regarding cell shape, cell volume, and atomic positions, using a conjugate gradient algorithm and converging forces to 10 μeV Å$^{-1}$ and electronic energies to 1 μeV. The initial crystal structures were obtained from the inorganic crystal structure database [58] (*RE*H$_3$: YH$_3$ *P*$\bar{3}$*c*1 deposition number 154809; *RE*$_2$O$_3$: Lu$_2$O$_3$ *Ia*$\bar{3}$ 194467; *RE*HO: LaHO *P*4/*nmm* 48122, YDO *Pnma* 1903936, HoHO *F*$\bar{4}$3*m* 1993608, YOF *R*$\bar{3}$*m* 184004, LiH *Fm*$\bar{3}$*m* 61751, LiF *Fm*$\bar{3}$*m* 41409, CaH$_2$ *Pnma* 261185, CaO *Fm*$\bar{3}$*m* 163628). A- and B-type sesquioxides generally yielded higher absolute energies, which is why only the C-type structure was regarded. The total energies derived from the structure optimizations were used to calculate free reaction enthalpies $\Delta_r G$ and differences in free reaction enthalpies $\Delta\Delta_r G$.

## 3. Results and Discussion

### 3.1. DyHO and ErHO

Both samples were obtained as light-gray powders. That of Er showed a high yield in ErHO, while of the dysprosium sample contained significant amounts of Dy$_2$O$_3$. The latter could not be decreased by excess DyH$_3$ and a second annealing; the present composition is therefore attributed to the thermal instability of DyHO, i.e., a partial decomposition to Dy$_2$O$_3$ and DyH$_3$ at the reaction temperature. This was similarly observed for HoHO [42] and is insignificant for ErHO. The temperature stability of the *RE*HO compounds thus increases with decreasing cationic radius, which is opposed to the trends predicted by DFT methods (see also Section 3.4 and [42]).

The NPD analysis confirmed the anion-ordered crystal structure previously observed for HoHO [42] (*F*$\bar{4}$3*m*, half-Heusler LiAlSi type; Figures 2–4, Tables 1 and 2). The high background in the diffraction patterns is attributed to the large incoherent scattering contribution of H [52], and the progression of intensities in the dysprosium sample results from the high neutron absorption of Dy and the Debye-Scherrer diffraction geometry. The atomic distances and isotropic displacement factors agree with those in trigonal *RE*H$_3$ (Dy–H: 2.08–2.46 Å, Er–H: 2.05–2.43 Å [59]), as well as cubic Dy$_2$O$_3$ (Dy–O: 2.25–2.35 Å, $B_{iso}$(Dy) = 0.24–0.28 Å$^2$, $B_{iso}$(O) = 0.40 Å$^2$) [60] and Er$_2$O$_3$ compounds (Er–O: 2.24–2.33 Å, $B_{iso}$(Er) = 0.1–1.3 Å$^2$, $B_{iso}$(O) = 0.3–2.3 Å$^2$ [61,62]), and the homologous hydride oxide HoHO ($B_{iso}$(Ho) = 0.3 Å$^2$, $B_{iso}$(H) = 1.9 Å$^2$, $B_{iso}$(O) = 0.6 Å$^2$) [42]. Further details on the crystal structures can be obtained from the Inorganic Crystal Structure Database (ICSD) [58], quoting the deposition numbers CSD-2082352 (DyHO) and CSD-2082354 (ErHO). The neutron diffraction data and the corresponding GSAS-II .gpx files are provided as supplementary material.

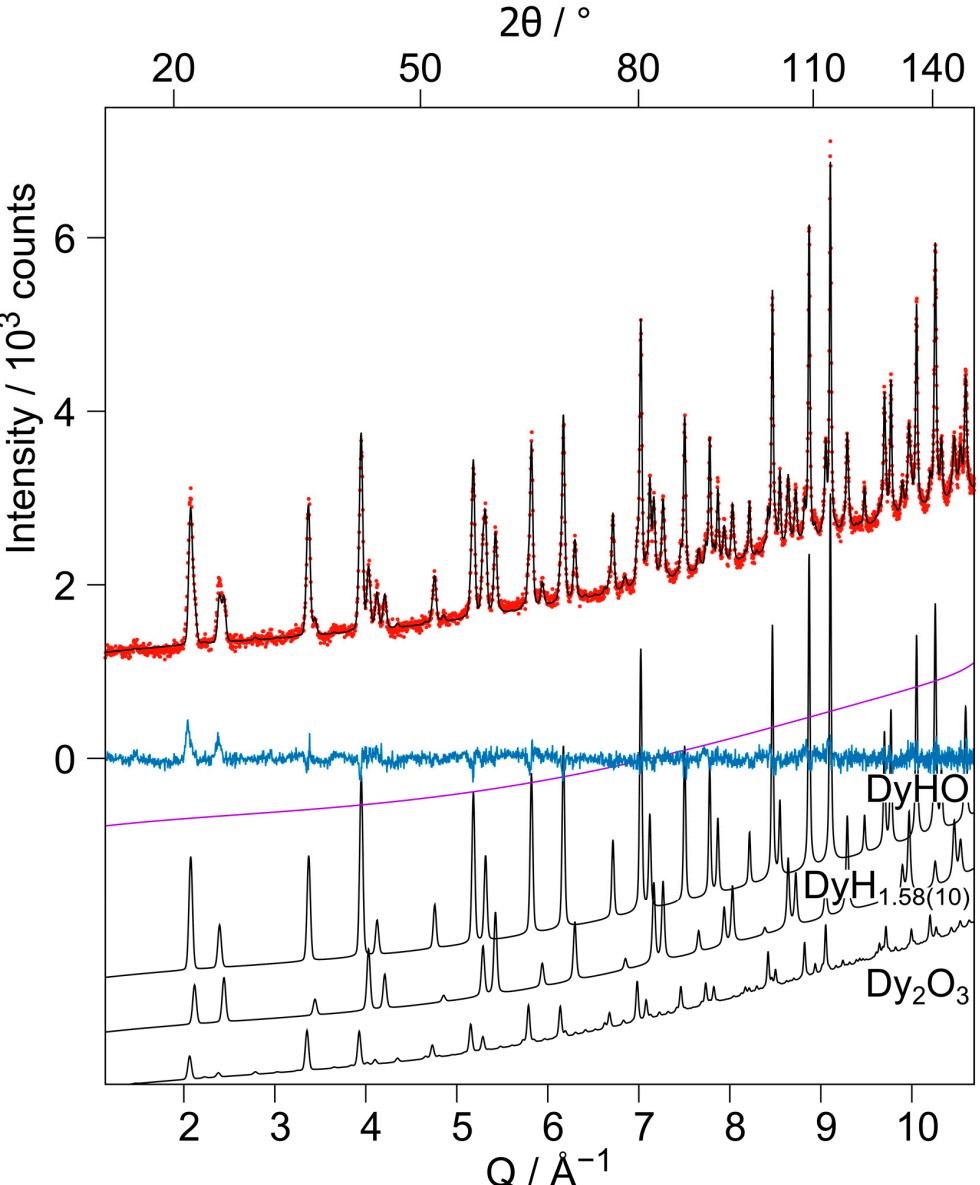

**Figure 2.** Rietveld refinement of the crystal structures of DyHO based on neutron diffraction data at ambient temperature ($\lambda$ = 1.155 Å) and contributions of the included crystal structures (bottom). $R_{wp}$ = 2.89%, *GoF* = 1.4. Phase information: DyHO $F\bar{4}3m$ (59.0(6) wt%, $R_{F^2}$ = 11.7%, *a* = 5.30945(10) Å), DyH$_{1.58(10)}$ $Fm\bar{3}m$ (24.2(4) wt%, *a* = 5.20261(20) Å), Dy$_2$O$_3$ $Ia\bar{3}$ (16.8(6) wt%, *a* = 10.6795(7) Å). Red dots: measurement *meas*, black curve: calculation *calc*, blue curve: difference *meas-calc*, purple curve: background.

**Table 1.** The crystal structure parameters of DyHO and ErHO based on neutron diffraction data at ambient temperature [1].

| | | | | | $B_{iso}$/Å$^2$ | |
|---|---|---|---|---|---|---|
| **Site** | **Wyck.** | *x* | *y* | *z* | **DyHO** | **ErHO** |
| Dy/Er | 4*a* | 0 | 0 | 0 | 0.316(11) | 0.103(11) |
| H | 4*c* | 1/4 | 1/4 | 1/4 | 2.46(16) | 1.22(2) |
| O | 4*d* | 3/4 | 3/4 | 3/4 | 0.84(5) | 0.178(16) |

[1] DyHO: $F\bar{4}3m$, *a* = 5.30945(10) Å. ErHO: $F\bar{4}3m$, *a* = 5.24615(7) Å.

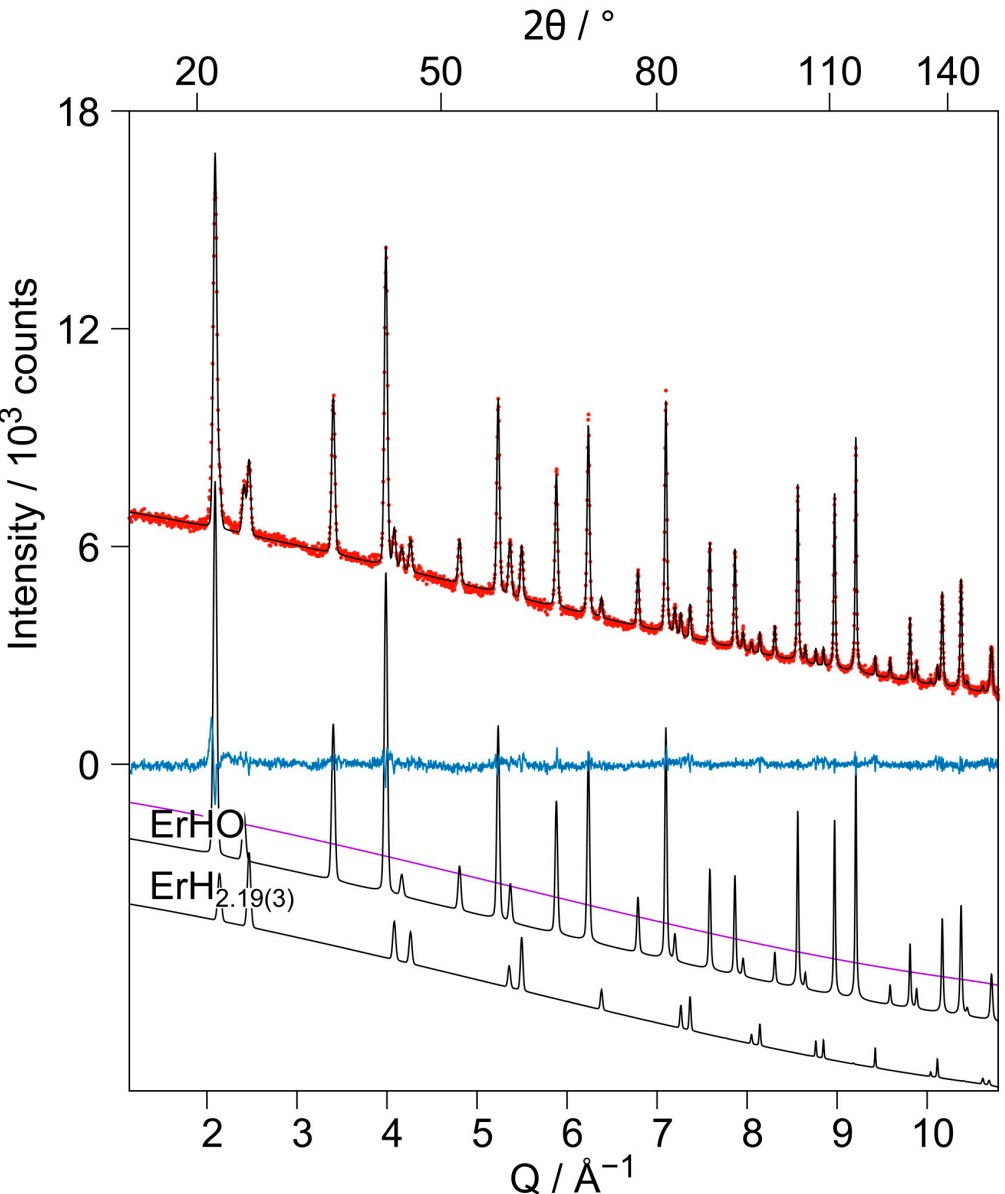

**Figure 3.** Rietveld refinement of the crystal structures of ErHO based on neutron diffraction data at ambient temperature ($\lambda = 1.155$ Å) and contributions of the included crystal structures (bottom). $R_{wp} = 2.50\%$, $GoF = 1.6$. Phase information: ErHO $F\bar{4}3m$ (81.2(3) wt%), $R_{F^2} = 6.7\%$, $a = 5.24615(7)$ Å), ErH$_{2.25(3)}$ $Fm\bar{3}m$ (18.8(3) wt%), $a = 5.12594(21)$ Å). Red dots: measurement *meas*, black curve: calculation *calc*, blue curve: difference *meas-calc*, purple curve: background.

**Table 2.** The interatomic distances in DyHO and ErHO based on neutron diffraction data at ambient temperature.

| | | Distance/Å | |
|---|---|---|---|
| **Atom 1** | **Atom 2** | **DyHO** | **ErHO** |
| Dy/Er | H O | 2.29906(4) | 2.27165(3) |
| H | O | 2.65473(5) | 2.62308(4) |
| H O | H O | 3.75435(7) | 3.70959(5) |

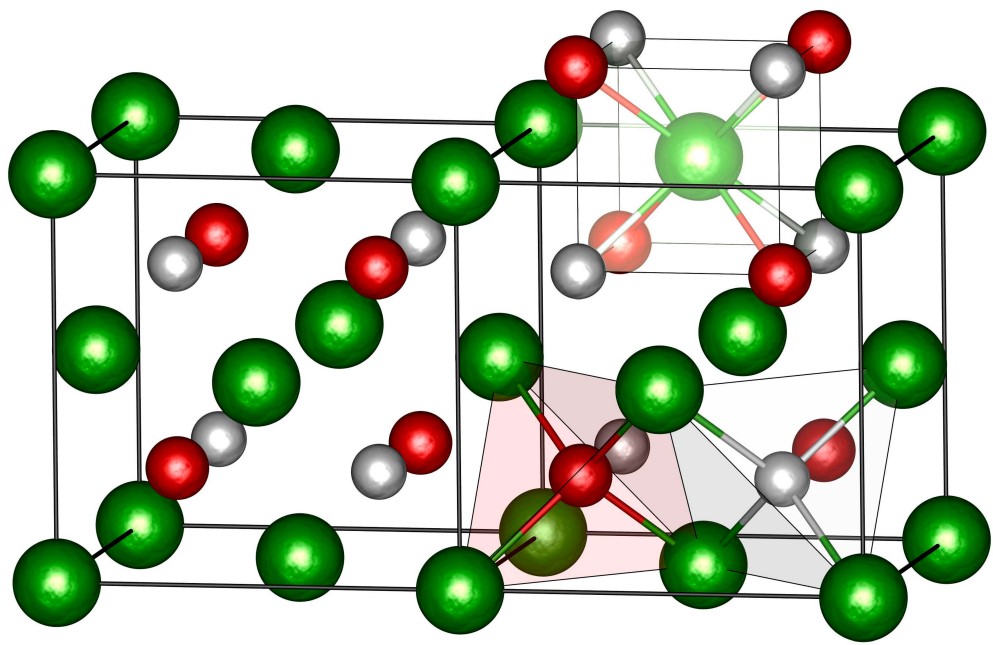

**Figure 4.** The crystal structure and coordination polyhedra of DyHO and ErHO; two unit cells are shown. Color code: Dy/Er, green; H, white; O, red.

### *3.2. LuHO and LuDO*

### 3.2.1. Cubic Half-Heusler LiAlSi Type ($F\bar{4}3m$)

The reaction of lutetium oxide and lutetium hydride produced a gray powder whose XRPD analysis showed a typical fcc-type reflection pattern. It could be attributed to a cubic LuHO phase, which has a significantly larger lattice parameter than the corresponding dihydride (5.171591(13) Å instead of 5.0368(5) Å; Figure A1, Tables A1 and A2). The Rietveld refinement with the anion-ordered structure model produced a slightly superior fit ($\Delta R_\mathrm{I}$ = 0.08%), and the lattice parameter was nearly identical with that extrapolated from anion-ordered *RE*HO compounds (see also Section 3.3, Table 3). Therefore, we assumed cubic LuHO to crystallize in the anion-ordered structure model with space group $F\bar{4}3m$. Additionally, small amounts of a second LuHO polymorph (orthorhombic anti-LiMgN type; details see below) were formed (4.24(13) wt%), that can be distinguished from lutetium oxide especially at high glancing angles.

**Table 3.** The interatomic distances in orthorhombic LuDO based on neutron diffraction data and orthorhombic LuHO based on X-ray diffraction data at ambient temperatures. Mean values are italicized.

| | | | Distance/Å$^2$ | |
|---|---|---|---|---|
| **Atom 1** | **Atom 2** | **Quantity** | **LuDO** | **LuHO** |
| Lu | D/H | 1 | 2.232(2) | 2.140(2) [2] |
| | | 1 | 2.307(17) | 2.413(2) [2] |
| | | 2 | 2.344(10) | 2.396(13) [2] |
| | | | *2.307* | *2.336* [2] |
| Lu | O | 1 | 2.134(19) | 2.11(3) |
| | | 1 | 2.188(17) | 2.22(3) |
| | | 2 | 2.226(7) | 2.222(14) |
| | | | *2.194* | *2.194* |
| D/H | D/H | 2 | 2.44(3) | 2.46 [2] |
| D/H | O | 1 | 2.468(15) | 2.43(3) [2] |
| | | 2 | 2.60(3) | 2.59(2) [2] |
| | | 1 | 2.747(15) | 2.78(3) [2] |
| | | | *2.604* | *2.598* [2] |
| O | O | 2 | 2.72(3) | 2.81(5) |

[2] The atomic parameters of H were set equal to those refined on the deuteride; the reported values are thus only estimates based on the atomic parameters of Lu and O and the lattice parameters of LuHO.

### 3.2.2. Orthorhombic Anti-LiMgN Type (*Pnma*)

The gray powder obtained from the reaction of lutetium hydride and calcium oxide showed an XRPD reflection pattern similar to orthorhombic YHO with space group *Pnma* in addition to significant amounts of $CaH_2$. A Rietveld refinement starting from the structure model of anti-LiMgN type LuHO resulted in a good fit. As the reflection patterns of orthorhombic *RE*HO and cubic bixbyite type $RE_2O_3$ are very similar, an unambiguous distinction between both structure models is important. An easy yet indicative parameter is the lattice parameter distortion from the cubic $CaF_2$ parent cell, which can be derived from the $CaF_2$ symmetry tree [42] and is equal to 1 for $Lu_2O_3$. The distortion was significant: $a/2b = 0.999986(14)$ (the $a/2b$ value does not differ significantly from 1 here; a refinement in the tetragonal LaHO type structure, however, produced inferior fit results ($\Delta R_{wp} = 2.7\%$)), $a/\sqrt{2}c = 0.99966(4)$, $\sqrt{2}b/c = 0.99968(5)$ (LuDO: 1.001801(6), 0.99384(9), 0.99206(8); YDO: 1.003186(5), 1.001119(2), 0.997939(3), YHO: 1.002807(13), 1.000817(17), 0.998015(3) [41]); ergo, orthorhombic LuHO was indeed formed. Similar approaches for ErHO and YbHO were successful for the former, producing cubic ErHO, while $Yb_2O_3$ was formed instead of the latter, similar to our observations for HoHO [42]. The LuHO sample's crystallinity could not be increased by subsequent annealing at higher temperatures (950 K), but instead resulted in the formation of lutetium oxide.

To determine the hydrogen positions, we performed NPD measurements on a deuterated sample LuDO. Deuterium has a smaller incoherent scattering contribution than protium, which results in a superior signal-to-noise ratio [52]. The Rietveld refinement confirmed the orthorhombic anti-LiMgN structure type (Figures 5 and 6, Tables 3 and 4). The bond lengths and isotropic displacement parameters were similar to those in isotypic YDO (Y–D: 2.30–2.43 Å, $B_{iso}(Y) = 0.20$ Å$^2$, $B_{iso}(D) = 1.67$ Å$^2$, $B_{iso}(O) = 0.24$ Å$^2$ [41]), trigonal $LuH_3$ (Lu–H: 2.02–2.39 Å, derived coordinates [59]), and bixbyite type $Lu_2O_3$ (Lu–O: 2.20–2.29 Å, $B_{iso}(Lu)$: 0.31–0.34 Å$^2$, $B_{iso}(O) = 0.47$ Å$^2$ [63]). The D–O distances were larger and the D–D and O–O distances smaller than in the cubic polymorph (Table A2). They increased in order with their anionic charge (D–D < D–O < O–O), and the large D–O distance supported the picture of an anionic nature of deuterium. The coordination polyhedron of Lu represented a distorted tetragonal prism with point symmetry *m*, while those of D and O were distorted tetrahedra. The isotropic displacement factor of lutetium did not differ significantly from zero. The tilt of the Lu coordination polyhedra toward the crystallographic *a* axis (6.1(5)°) was similar to that observed in YDO (5.0(1)°) and larger than those in tetragonal LaHO (2.803(3)° [36]) and NdHO (4.049(2)° [37]), as well as monoclinic LaDO (1.8(3)° [34]) (tilt toward *c*, due to interchanged *a* and *c* axes). The crystal structure parameters of LuHO obtained from XRPD only differed in their lattice parameters (Figure A2, Tables 3–5).

**Table 4.** The crystal structure parameters of orthorhombic LuHO based on X-ray diffraction data at ambient temperature [3].

| Site | Wyck. | $x$ | $y$ | $z$ | $B_{iso}$/Å$^2$ |
|---|---|---|---|---|---|
| Lu | 4*c* | 0.3629(3) | 1/4 | 0.2988(4) | $B_{iso}$(Lu, LuDO) + $B_{overall}$ [4] |
| O | 4*c* | 0.642(3) | 1/4 | 0.461(5) | $B_{iso}$(O, LuDO) + $B_{overall}$ [4] |

[3] *Pnma*, $a = 7.3493(7)$ Å, $b = 3.6747(4)$ Å, $c = 5.1985(3)$ Å; [4] $B_{overall} = -2.13$ Å$^2$.

**Table 5.** The crystal structure parameters of orthorhombic LuDO based on neutron diffraction data at ambient temperature [5].

| Site | Wyck. | $x$ | $y$ | $z$ | $B_{iso}$/Å$^5$ |
|---|---|---|---|---|---|
| Lu | 4*c* | 0.3740(17) | 1/4 | 0.2875(12) | −0.13(7) |
| D | 4*c* | 0.111(3) | 1/4 | 0.5053(21) | 1.63(17) |
| O | 4*c* | 0.636(3) | 1/4 | 0.4678(18) | 0.85(17) |

[5] *Pnma*, $a = 7.3116(16)$ Å, $b = 3.6492(8)$ Å, $c = 5.2021(7)$ Å.

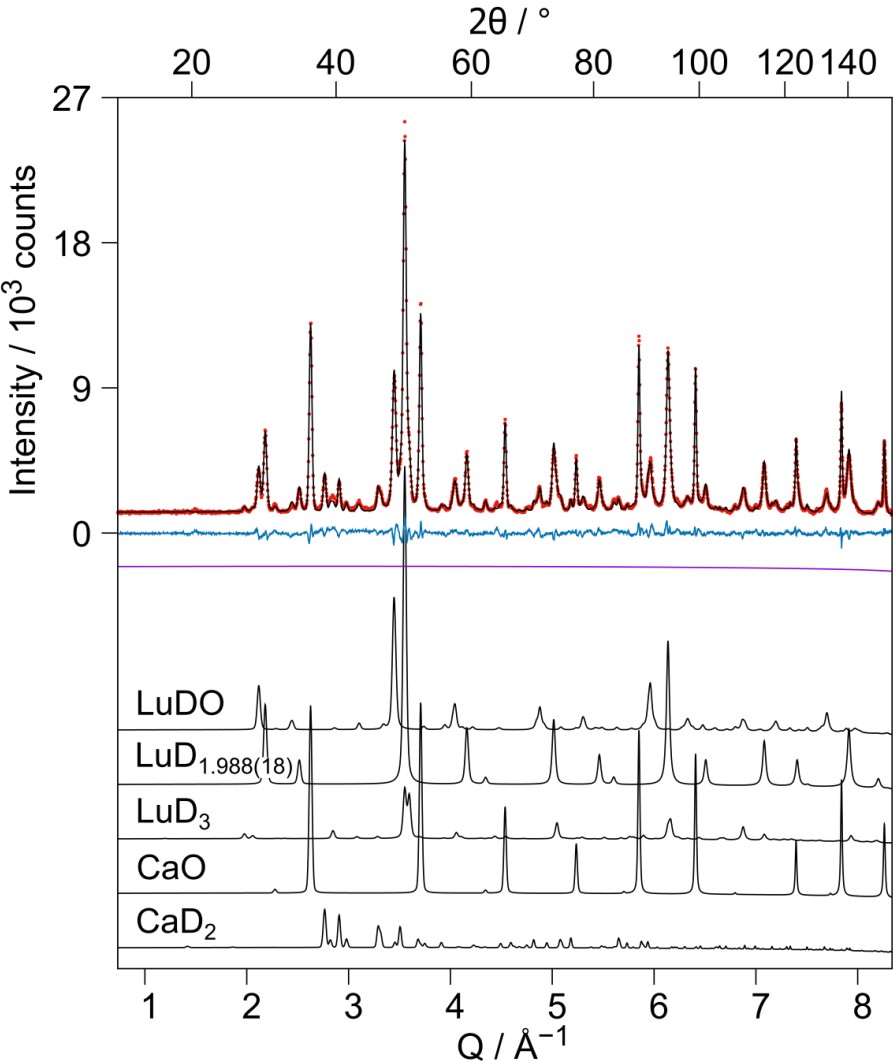

**Figure 5.** Rietveld refinement of the crystal structure of LuDO based on neutron diffraction data ($\lambda$ = 1.494 Å) and contributions of the included crystal structures (bottom). $R_{wp}$ = 5.18%, *GoF* = 2.5. Phase information: LuDO *Pnma* (24.8(3) wt%), $R_{F^2}$ = 4.8%, *a* = 7.3116(16) Å, *b* = 3.6492(8) Å, *c* = 5.2021(7) Å), LuD$_{1.988(18)}$ *Fm$\bar{3}$m* (42.2(4) wt%), $R_{F^2}$ = 2.6%, *a* = 5.02497(13) Å), LuD$_3$ *P$\bar{3}$c1* (8.47(19) wt%), *a* = 6.1542(18) Å, *c* = 6.4064(10) Å), CaO *Fm$\bar{3}$m* (19.73(11) wt%), *a* = 4.81183(6) Å), CaD$_2$ *Pnma* (4.84(11) wt%), *a* = 5.9497(7) Å, *b* = 3.5987(5) Å, *c* = 6.8046(9) Å. Red dots: measurement *meas*, black curve: calculation *calc*, blue curve: difference *meas-calc*, purple curve: background.

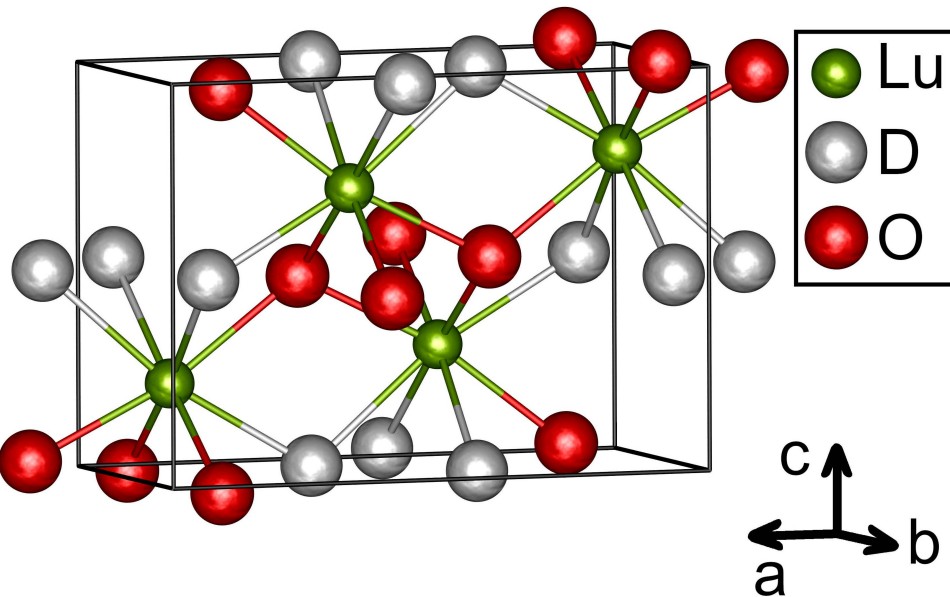

**Figure 6.** The crystal structure of orthorhombic LuDO.

In contrast to the tetragonal LaHO type [12], the formation of the orthorhombic anti-LiMgN type and its monoclinically distorted variety could not be correlated with the cationic radii. The latter were only observed for cations with empty (Y, La) or fully occupied *f* orbitals (Lu), but not for cations with similar size. We can only assume that their formation might therefore depend on the polarizability of the cations.

We additionally obtained the crystal structure parameters of the secondary phase LuD$_{1.985(19)}$ (Figure 5, Tables A3 and A4), which is the first reported NPD analysis of a binary lutetium deuteride. Further details on all presented crystal structures can be received from the Inorganic Crystal Structure Database (ICSD) [58], quoting the deposition numbers CSD-2082351 (orthorhombic LuDO), CSD-2082353 (orthorhombic LuHO), CSD-2082355 (cubic LuHO), and CSD-2082356 (LuD$_{1.985(19)}$). The neutron diffraction data and the corresponding GSAS-II .gpx files are provided as supplementary material.

*3.3. A Comparison of Cubic Rare Earth Hydride Oxides*

The differentiation of cubic anion-ordered ($F\bar{4}3m$) and disordered ($Fm\bar{3}m$) rare earth hydride oxides by standard laboratory methods, i.e., without using neutrons, remains a challenge. While intensities in neutron diffraction experiments on $^1$H compounds exhibit very high contrast, that in X-ray diffraction measurements is almost negligible and therefore not unambiguous [42]. $F\bar{4}3m$ and $Fm\bar{3}m$ belong to the same Laue group, and therefore electron diffraction is not suitable as a differentiation tool either. Interpretation of NMR spectra is probably not indicative, as the coupling of core spin and *f* electrons leads to broad signals. Vibrational spectroscopy might be suitable and was successfully conducted on NdHO before [37], albeit our efforts on Raman spectroscopy at 532 nm excitation wavelength remain fruitless because only very broad signals with poor signal-to-noise ratios due to strong fluorescence are produced. However, for anion-ordered and disordered holmium hydride oxide, a significant discrepancy in their lattice parameters was reported ($\Delta a$ = 0.03264(3) Å, $\Delta a / a_{F\bar{4}3m}$ = 0.6464(6)%) [42]. We therefore performed the same comparison for all hitherto reported bulk *RE*HO compounds (Figure 7, Table 3). The lattice parameters of anion-ordered DyHO, HoHO, and ErHO can be correlated with the tabulated radii of the eightfold coordinated $RE^{3+}$ ions [64], resulting in a trend line with high significance.

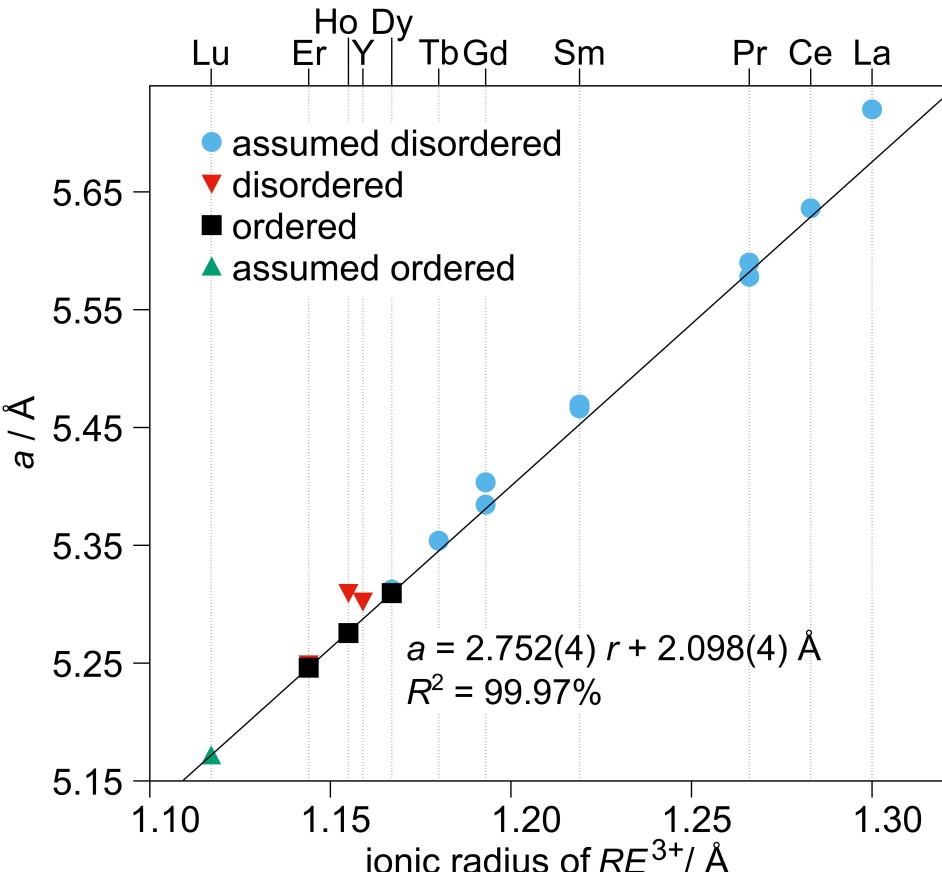

**Figure 7.** Correlation of lattice parameters of cubic rare earth hydride oxides with anion-ordered or -disordered structure verified by neutron diffraction, or yet unknown and assumed ordered or disordered anionic substructure with the ionic radius of the eightfold coordinated $RE^{3+}$ ion [64]. The error bars are smaller than the displayed symbols. Black line: linear regression of anion-ordered $RE$HO compounds DyHO, HoHO, and ErHO. Values for $RE$ = La, Ce, Pr [31,32], Sm [12,40], Gd [12,20], Tb, Dy [12], Ho [12,42], Er [7,12] were obtained from the literature.

While some compounds show a large discrepancy from this trend (LaHO, GdHO, YHO, HoHO), others show only small or negligible deviations (CeHO, PrHO, SmHO, one GdHO, TbHO, DyHO, ErHO, LuHO). Especially the negligible difference of the verified-disordered ErHO compound is interesting, as it disproves the initial hypothesis of the lattice parameter being indicative for the anion ordering. As the relative deviations $\Delta a / a_{F\bar{4}3m}$ are not correlated with the cationic radius, as would be expected for a series of ionic rare earth compounds, and rather different values were reported for PrHO and GdHO (though from different groups), the reason for the discrepancies must be correlated with the compound's stoichiometry. As reviewed in the introduction, several indications for a phase width of ternary $RE$–H–O compounds were reported, which means a distinction between compounds $RE$H$_{1+2x}$O$_{1-x}$, with both $x < 0$ (O-rich hydride oxides) and $x > 0$ (H-rich hydride oxides) and $RE$HO ($x = 0$) is necessary. The fact that all significant deviations in Figure 7 and Table 6 are positive can thus be explained by crystal chemical arguments. In comparison to $x = 0$, the distance of high charged oxide anions decreases for $x < 0$, and additional crystallographic sites, most probably the octahedral interstice, have to be occupied for $x > 0$. Both factors result in an expansion of the unit cell volume. Therefore, the lattice parameter may thus only be a criterion to differentiate $RE$HO and $RE$H$_{1+2x}$O$_{1-x}$ compounds.

**Table 6.** Lattice parameters of cubic hydride oxides and their relative deviations from the values of anion ordered phases with space group $F\bar{4}3m$. In cases where the anion-ordered compound is not known, extrapolated values were employed (see Figure 7).

| RE | $a$(REHO, $F\bar{4}3m$)/Å | $a$(REHO $Fm\bar{3}m$)/Å | $a$(REHO, Order Assumed) | $\Delta a/a_{F\bar{4}3m}$/% |
|---|---|---|---|---|
| Y | | 5.3023(7) [41] | | 0.339 |
| La | | | 5.720 [31,32] [6] | 1.048 |
| Ce | | | 5.636 [31,32] [6] | 0.368 |
| Pr | | | 5.578 [31] [6] | 0.144 |
| Pr | | | 5.590 [32] [6] | 0.360 |
| Sm | | | 5.46619(4) [12] [6] | 0.398 |
| Sm | | | 5.46953(6) [40] [6] | 0.460 |
| Gd | | | 5.38450(4) [20] [6] | 0.175 |
| Gd | | | 5.4034834(19) [12] [6] | 0.528 |
| Tb | | | 5.353841(4) [12] [6] | 0.251 |
| Dy | 5.30945(10) | | 5.312418(7) [12] [6] | 0.056(2) |
| Ho | 5.27550(13) [42] | 5.3096(1) [12] | | 0.6464(6) |
| Er | 5.24615(7) | 5.2490(1) [7] | 5.247284(14) [12] [6] | 0.0543(6) [7] |
| Lu | | | 5.171591(13) | −0.013 |

[6] Only XRPD data, ordered modification not tested; [7] calculated using the value of the verified anion-disordered compound.

### 3.4. A Comparison of Synthesis Routes by DFT Calculations

A number of different synthesis approaches were reported for ternary rare earth hydride oxides, usually starting from $RE_2O_3$, $REH_3$, or $RE$OF [7,12,20,31,32,36,37,40–42]. Their success depends on $RE$ and is governed by the thermodynamic stability of the reaction products compared with those of the educts. A measure for this is the free reaction enthalpy $\Delta_r G$ that can easily be obtained from quantum mechanical calculations. For the general reaction scheme $a\,A + b\,B \to c\,C + d\,D$, $\Delta_r G$ equals the difference of the total energies $E$ derived from the Schrödinger (or Kohn–Sham) equations: $\Delta_r G = c\,E(C) + d\,E(D) - a\,E(A) - b\,E(B)$. Negative values thus indicate that the reaction product (or mixture of reaction products) is thermodynamically more stable than the mixture of educts. The $\Delta_r G$ values for the following chemical equations were calculated:

$$RE_2O_3 + CaH_2 \to 2\,RE\text{HO} + CaO \tag{1}$$

$$REH_3 + CaO \to RE\text{HO} + CaH_2 \tag{2}$$

$$RE_2O_3 + REH_3 \to 3\,RE\text{HO} \tag{3}$$

$$RE\text{OF} + LiH \to RE\text{HO} + LiF \tag{4}$$

The results are presented in Figure 8 and Tables A5 and A6.

For the reaction (1), $\Delta_r G$ decreased with increasing $r(RE^{3+})$, i.e., the stability of $RE$HO toward the oxide increased with $r$. Positive values were observed for the smallest representatives (Ho–Lu), which explains the literature reports of failed attempts via this route for TmHO, YbHO, and LuHO [12]. For the reaction (2), the trend was inverse—the stability of the hydride oxide toward the hydride increased with decreasing $r(RE^{3+})$. The third reaction (3) combines both aforementioned trends, with the stability of the oxide superseding the instability of the hydride, i.e., the stability of the hydride oxide decreases along with $r(RE^{3+})$. The latter might result from a decreasing softness of cations. The fourth reaction's (4) trend is much less pronounced and showed a small preference for the smaller rare earth elements.

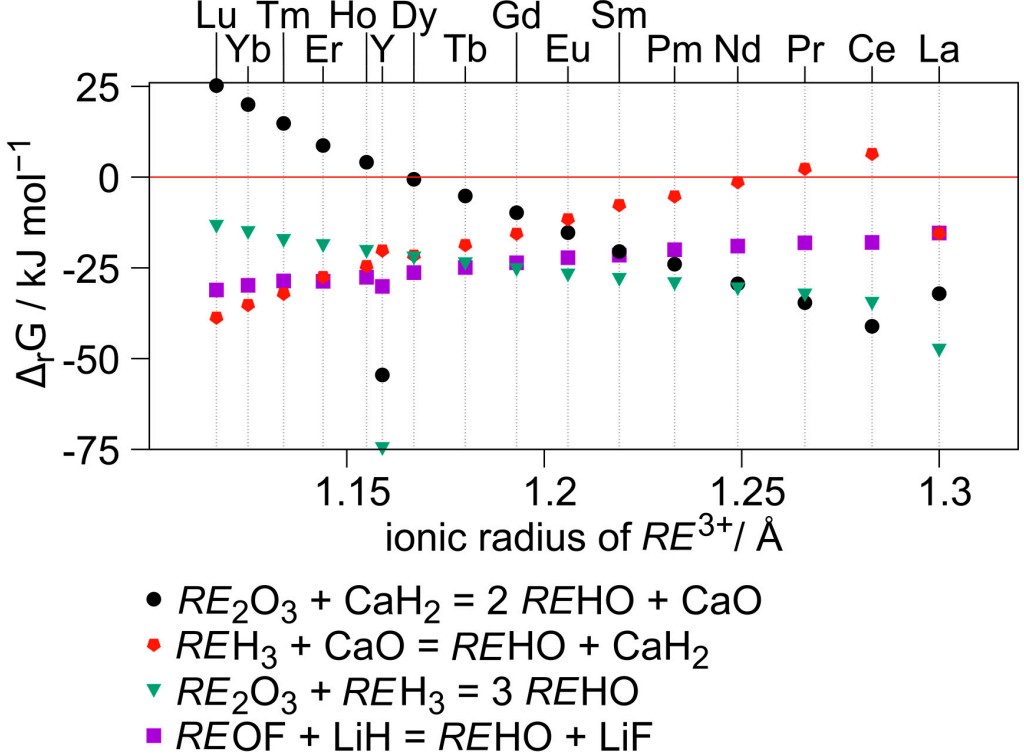

**Figure 8.** Plot of the free reaction enthalpies $\Delta_r G$ for different synthesis approaches against the ionic radii of eightfold coordinated $RE^{3+}$ ions [64].

A previous study observed similar trends for reaction (1) [65]. Instead of Equation (3), a reaction starting from the sesquioxide, rare earth metal, and gaseous hydrogen was calculated: $RE_2O_3 + RE + 1.5\,H_2 \rightarrow 3\,REHO$, which showed an inverse trend to our observations with exception of LaHO. The discrepancy might have resulted from the different educts ($RE + H_2$ instead of $REH_3$) or the different computational method.

A second important factor in the synthesis of $REHO$ compounds are competing side reactions with greater thermodynamic potential difference. To compare different reactions, we calculated the difference in free reaction enthalpies $\Delta\Delta_r G$. For the system $RE_2O_3/CaH_2$, the metathesis as side reaction was therefore compared with the formation of $REHO$:

$$RE_2O_3 + 3\,CaH_2 \rightarrow 2\,REH_3 + 3\,CaO \tag{5}$$

$$RE_2O_3 + 3\,CaH_2 \rightarrow 2\,REHO + CaO + 2\,CaH_2 \tag{6}$$

$\Delta\Delta_r G$ is calculated as difference of the respective free reaction enthalpies: $\Delta\Delta_r G(RE_2O_3/CaH_2) = \Delta_r G(6) - \Delta_r G(5)$. For the system $REH_3/CaO$, the following equations are regarded:

$$2\,REH_3 + 3\,CaO \rightarrow RE_2O_3 + 3\,CaH_2 \tag{7}$$

$$2\,REH_3 + 3\,CaO \rightarrow 2\,REHO + CaH_2 + 2\,CaO \tag{8}$$

In addition, $\Delta\Delta_r G(REH_3/CaO) = \Delta_r G(8) - \Delta_r G(7)$. Finally, the equations for the system $REOF/LiH$:

$$3\,REOF + 3\,LiH \rightarrow RE_2O_3 + REH_3 + 3\,LiF \tag{9}$$

$$3\,REOF + 3\,LiH \rightarrow 3\,REHO + 3\,LiF \tag{10}$$

In addition, $\Delta\Delta_r G(REOF/LiH) = \Delta_r G(10) - \Delta_r G(9)$. The results are presented in Figure 9 and Table A7. In all cases, negative values of $\Delta\Delta_r G$ indicate the favored formation of the hydride oxide.

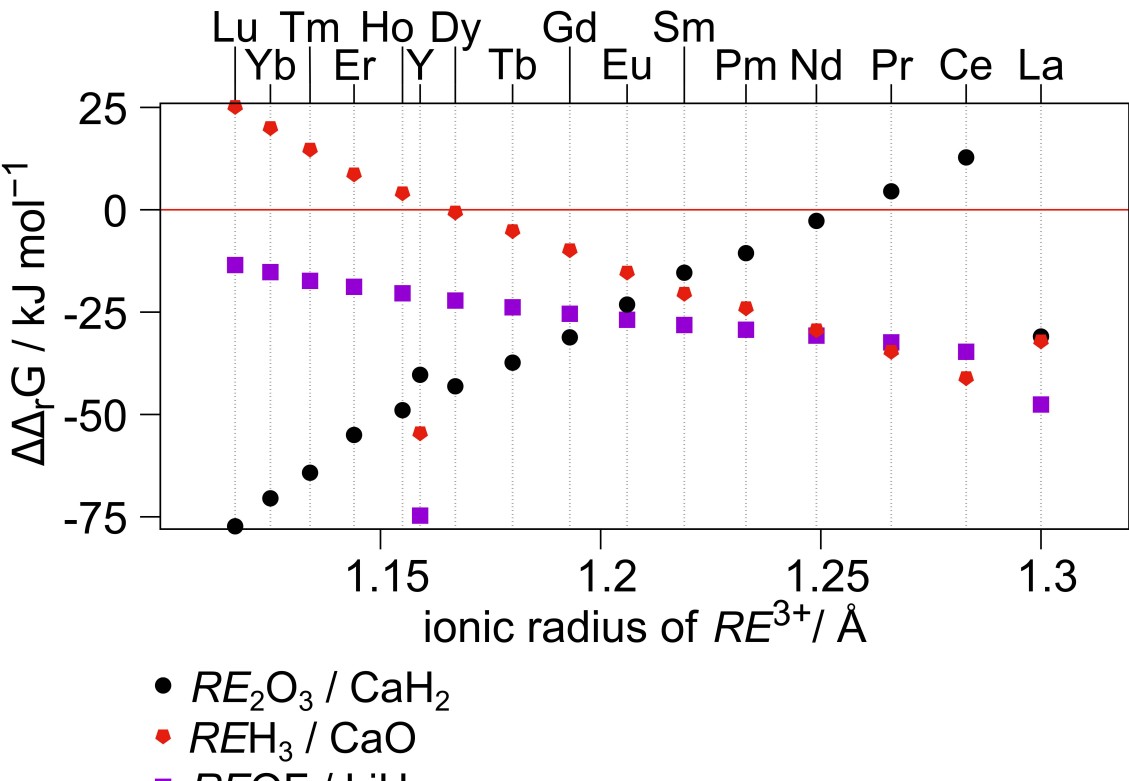

**Figure 9.** Differences of the free reaction enthalpies $\Delta\Delta_r G$ of the three *RE*HO synthesis approaches using alkaline and alkaline earth compounds and their respective side reactions against the ionic radii of eightfold coordinated $RE^{3+}$ ions [64]. Negative values: formation of *RE*HO is favored.

For the system $RE_2O_3/CaH_2$, the formation of *RE*HO was more favored for the smaller $r(RE^{3+})$ and the metathesis only for Ce and Pr. In the system $REH_3/CaO$, the formation of *RE*HO was disfavored for the smaller lanthanides (Ho–Lu). These findings explain the reaction of $REH_3$ and CaO to $RE_2O_3$ for $RE$ = Ho and Yb [42] but seem to contradict our observations for LuHO. However, annealing at higher temperatures resulted in further reaction with CaO to lutetium oxide (see also the Results section—LuHO), thus suggesting the formation of LuHO to be kinetically controlled. The third system $REOF/LiH$ only produced negative values with increasing magnitude for the larger lanthanides and yttrium.

When the trends in $\Delta_r G$ and $\Delta\Delta_r G$ are compared, general guidelines for a successful *RE*HO synthesis can be extracted and their thermal stability extrapolated. For the smaller rare earth elements, the binary oxides are more stable than the hydrides, as $\Delta_r G$ ($RE_2O_3$ + $CaH_2$) and $\Delta\Delta_r G(REH_3/CaO) > 0$; similarly, the binary hydrides are more stable than the oxides for larger $RE$, as the magnitude of $\Delta_r G(REH_3 + CaO)$ and $\Delta\Delta_r G(RE_2O_3/CaH_2)$ decrease with increasing $r$ and become positive for cerium and praseodymium, with lanthanum representing an exception. The hydride oxides are always more stable than the binary compounds, but with decreasing magnitude with smaller cationic radius ($\Delta_r G(RE_2O_3$ + $REH_3)$). As the decreasing stability of the binary hydrides runs along with that of the hydride oxide, the synthesis starting from $REH_3$ and CaO appears promising at first due to the higher thermodynamic potential. It is however superseded by the stability of the oxide, as the $\Delta\Delta_r G$ plot shows.

The thermal stability of the *RE*HO compounds is furthermore important, as solid-state reactions demand high temperatures. A decomposition toward the binary compounds, which we previously observed for HoHO above 540 K [42], is more probable for a smaller $\Delta_r G$ value, i.e., for smaller $r$. The *RE*HO representatives of the smallest *RE* may therefore only be obtained by a kinetic reaction control, as described above for LuHO, or from the oxide fluorides by F–H metathesis. The experimentally observed thermal stability of the anion ordered cubic *RE*HO compounds (see Section 3.1) contradict the trend described by the DFT calculations and might indicate an insufficient description of the compounds by the employed pseudopotentials.

## 4. Conclusions

We prepared five new rare earth hydride oxides, DyHO, ErHO, LuDO, as well as cubic and orthorhombic LuHO, and investigated their structure by neutron and X-ray powder diffraction. DyHO, ErHO, and LuHO crystallized in the anion-ordered modification with space group $F\bar{4}3m$, similar to the previously reported HoHO [42]. LuHO and LuDO furthermore crystallized in the orthorhombic anti-LiMgN structure type. A comparison of the cubic rare earth hydride oxide's lattice parameters $a$ showed its sensitivity to differentiate $RE$HO and $REH_{1+2x}O_{1-x}$ compounds, whereas the values of the latter were significantly larger than those extrapolated from the former. This differentiation is especially important in the research of their ionic conductivity. The lattice parameter can, however, not be used to distinguish between the cubic anion-ordered and disordered modifications of $RE$HO. Our study thus stresses the importance of a thorough understanding of the anionic substructure in the system $RE$–H–O, which is more complex than previously assumed, though probably none of the reported substances is interesting for applications in ion conductors [7] or catalysts [12] due to the small cations with less efficient Coulomb shielding and thus small hydride ion diffusivity. Furthermore, DFT calculations supported our experimental findings: The compounds $RE$HO are more stable for the larger rare earth elements, whereas the sesquioxides become more stable than $RE$HO with decreasing cationic radius. For the smallest representatives including Lu, mild reaction conditions for the stabilization of kinetic products are thus necessary. Alternatively, a synthesis from the oxide fluorides $RE$OF might be successful for the thus far elusive Tm, Yb, and Sc compounds.

**Supplementary Materials:** The following are available online at https://www.mdpi.com/article/10.3390/cryst11070750/s1, Neutron diffraction data of DyHO, ErHO, and LuDO and the respective GSAS-II gpx-files.

**Author Contributions:** Conceptualization, methodology, validation, formal analysis, data curation, writing—original draft preparation, visualization, N.Z.; resources, project administration, supervision, H.K.; investigation, N.Z. and D.S.; funding acquisition, N.Z. and H.K.; writing—review and editing, N.Z., D.S., and H.K. All authors have read and agreed to the published version of the manuscript.

**Funding:** This research was funded by the Operational Programme of the European Regional Development Fund 2014–2020, project "In situ investigations on energy related materials" (Project 100357551). This measure is cofinanced by tax money on the basis of the budget adopted by the members of the Saxon Landtag. This work was funded by the Deutsche Forschungsgemeinschaft (grant 419433503 and INST 268/379/1 FUGG).

**Data Availability Statement:** Reported crystal structures are available from the inorganic crystal structure database ICSD [58] quoting the deposititon numbers CSD-2082352 (DyHO), -2082354 (ErHO), -2082355 (cubic LuHO), -2082351 (orthorhombic LuDO), -2082353 (orthorhombic LuHO), and -2082356 (LuD$_{1.985(19)}$).

**Acknowledgments:** Parts of this work are based on experiments performed at the Swiss spallation neutron source SINQ, Paul Scherrer Institute, Villigen, Switzerland. The PhD scholarship from Leipzig University (Doktorandenförderplatz) is greatly appreciated. We acknowledge support from Leipzig University for Open Access Publishing. We thank F. Gehlhaar (Leipzig University) for providing starting materials and H. Krautscheid (Leipzig University) for access to an X-ray powder

diffractometer. Computations for this work were performed with resources of the Leipzig University Computing Centre. We thank A. Rost (Leipzig University) for his support as software administrator.

**Conflicts of Interest:** The authors declare no conflict of interest.

**Appendix A**

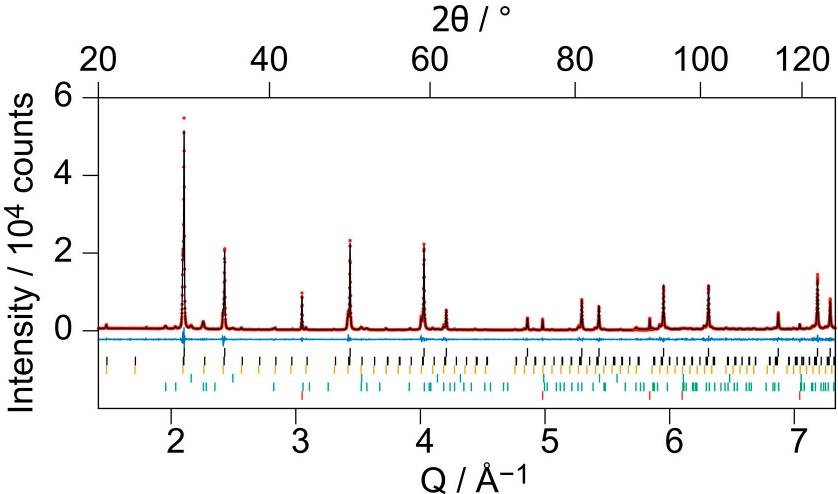

**Figure A1.** Rietveld refinement of the crystal structure of cubic LuHO obtained from $Lu_2O_3$ and $LuH_3$ on the basis of X-ray diffraction data at ambient temperature ($\lambda$: Cu-$K_{\alpha 1}$). $R_{wp}$ = 6.97%, *GoF* = 1.97. Bragg markers denote from top to bottom: LuHO $F\bar{4}3m$ (68.6(5) wt%, $R_I$ = 0.5%, $a$ = 5.171591(13) Å), LuHO *Pnma* (4.24(13) wt%), $Lu_2O_3$ *Ia*-3 (17.59(24) wt%, $a$ = 10.39605(11) Å), $LuH_2$ $Fm\bar{3}m$ (2.92(8) wt%, $a$ = 5.0367(5) Å), $LuH_3$ $P\bar{3}c1$ (6.60(11) wt%, 6.1727(4) Å, $c$ = 6.4257(6) Å), diamond (added to reduce X-ray absorption). Red dots: measurement *meas*, black curve: calculation *calc*, blue curve: difference *meas-calc*.

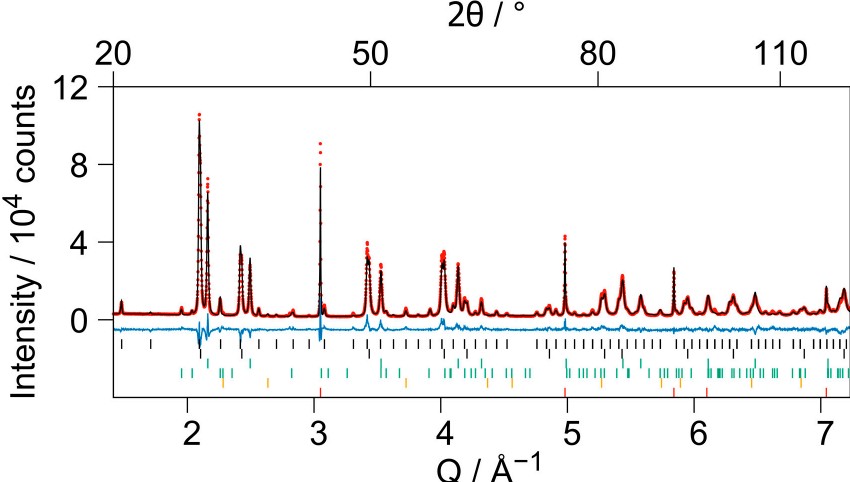

**Figure A2.** Rietveld refinement of the crystal structure of orthorhombic LuHO obtained from $LuH_3$ and CaO on the basis of X-ray diffraction data at ambient temperature ($\lambda$: Cu-$K_{\alpha 1}$). $R_{wp}$ = 10.1%, *GoF* = 7.3. Bragg markers denote from top to bottom: LuHO *Pnma* (31.3(4) wt%, $R_I$ = 7.9%, $a$ = 7.3493(7) Å, $b$ = 3.6747(4) Å, $c$ = 5.1985(3) Å), LuHO $F\bar{4}3m$ (35.8(4) wt%, $a$ = 5.17312(11) Å), $LuH_2$ $Fm\bar{3}m$ (27.32(17) wt%, $a$ = 5.03582(8) Å), $LuH_3$ $P\bar{3}c1$ (3.12(6) wt%, $a$ = 6.1689(6) Å, $c$ = 6.4317(10) Å), CaO $Fm\bar{3}m$ (2.4(3) wt%, $a$ = 4.7687(15) Å). Red dots: measurement *meas*, black curve: calculation *calc*, blue curve: difference *meas-calc*.

**Table A1.** The crystal structure parameters of cubic LuHO based on X-ray diffraction data at ambient temperature [A1].

| Site | Wyck. | *x* | *y* | *z* |
|------|-------|-----|-----|-----|
| Lu | 4*a* | 0 | 0 | 0 |
| H | 4*c* | 1/4 | 1/4 | 1/4 |
| O | 4*d* | 3/4 | 3/4 | 3/4 |

[A1] $F\bar{4}3m$, *a* = 5.171591(13) Å.

**Table A2.** The interatomic distances in cubic LuHO ($F\bar{4}3m$) up to 3 Å based on X-ray diffraction data at ambient temperature.

| Atom 1 | Atom 2 | Distance/Å |
|--------|--------|------------|
| Lu | H<br>O | 2.239365(6) |
| H | O | 2.585795(7) |
| H<br>O | H<br>O | 3.656867(9) |

**Table A3.** The crystal structure parameters of $LuD_{1.985(19)}$ based on neutron diffraction data at ambient temperature [A2].

| Site | Wyck. | *x* | *y* | *z* | SOF | $B_{iso}$/Å² |
|------|-------|-----|-----|-----|-----|--------------|
| Lu | 4*a* | 0 | 0 | 0 | 1 | 0.35(2) |
| D1 | 8*c* | 1/4 | 1/4 | 1/4 | 0.974(7) | 1.2(3) |
| D2 | 4*b* | 3/4 | 3/4 | 3/4 | 0.037(5) | $B_{iso}$(D1) |

[A2] $Fm\bar{3}m$, *a* = 5.02497(13) Å.

**Table A4.** The interatomic distances in $LuD_{1.985(19)}$ up to 3 Å based on neutron diffraction data at ambient temperature.

| Atom 1 | Atom 2 | Distance/Å |
|--------|--------|------------|
| Lu | D1 | 2.17588(6) |
| Lu | D2 | 2.51249(7) |
| D1 | D1 | 2.51249 (7) |
| D1 | D2 | 2.17588(6) |

**Table A5.** Total energies *E* per number of formula units *Z* from structure optimizations of the compounds $RE_2O_3$, $REH_3$, $REOF$, and $REHO$ [A3].

| | E/Z/MJ mol⁻¹ | | | | | |
|------|-----------|-----------|--------|----------------------|-------------------|-------------------|
| *RE* | $REH_3$ | $RE_2O_3$ | $REOF$ | $REHO\ F\bar{4}3m$ | $REHO\ P4/nmm$ | $REHO\ Pnma$ |
| Y | −1.837 | −4.363 | −2.401 | −2.092 | −2.089 | −2.090 |
| La | −1.672 | −4.046 | −2.246 | −1.919 | −1.921 | −1.922 |
| Ce | −1.638 | −3.925 | −2.187 | −1.865 | −1.865 | −1.866 |
| Pr | −1.643 | −3.950 | −2.197 | −1.875 | −1.874 | −1.875 |
| Nd | −1.647 | −3.970 | −2.203 | −1.883 | −1.881 | −1.882 |
| Pm | −1.650 | −3.990 | −2.209 | −1.890 | −1.888 | −1.889 |
| Sm | −1.649 | −3.996 | −2.209 | −1.891 | −1.889 | −1.890 |
| Eu | −1.651 | −4.013 | −2.214 | −1.897 | −1.894 | −1.896 |
| Gd | −1.653 | −4.031 | −2.219 | −1.903 | −1.900 | −1.902 |
| Tb | −1.653 | −4.041 | −2.220 | −1.906 | −1.903 | −1.904 |
| Dy | −1.651 | −4.048 | −2.220 | −1.907 | −1.904 | −1.906 |
| Ho | −1.650 | −4.055 | −2.220 | −1.908 | −1.905 | −1.907 |
| Er | −1.648 | −4.062 | −2.221 | −1.910 | −1.906 | −1.908 |
| Tm | −1.645 | −4.073 | −2.223 | −1.912 | −1.908 | −1.910 |
| Yb | −1.643 | −4.079 | −2.222 | −1.913 | −1.908 | −1.911 |
| Lu | −1.640 | −4.085 | −2.221 | −1.913 | −1.909 | −1.911 |

[A3] Total energies per number of formula units of LiH: −0.591 MJ mol⁻¹, LiF: −0.931 MJ mol⁻¹, $CaH_2$: −1.002 MJ mol⁻¹, CaO: −1.236 MJ mol⁻¹.

**Table A6.** Free reaction enthalpies $\Delta_r G$ for different synthesis approaches. Values were calculated for minimum values of $E(REHO)$ among the three modifications.

| | $\Delta_r G/\text{kJ mol}^{-1}$ | | | |
|---|---|---|---|---|
| *RE* | $RE_2O_3 + CaH_2 \rightarrow 2\,REHO + CaO$ | $REH_3 + CaO \rightarrow REHO + CaH_2$ | $RE_2O_3 + REH_3 \rightarrow 3\,REHO$ | $REOF + LiH \rightarrow REHO + LiF$ |
| Y | −54.5 | −20.2 | −74.7 | −30.1 |
| La | −32.1 | −15.5 | −47.6 | −15.4 |
| Ce | −41.1 | 6.4 | −34.7 | −18.0 |
| Pr | −34.6 | 2.3 | −32.4 | −18.1 |
| Nd | −29.4 | −1.4 | −30.7 | −19.0 |
| Pm | −24.0 | −5.3 | −29.3 | −20.0 |
| Sm | −20.5 | −7.7 | −28.1 | −21.5 |
| Eu | −15.3 | −11.6 | −26.9 | −22.2 |
| Gd | −9.8 | −15.6 | −25.4 | −23.6 |
| Tb | −5.2 | −18.7 | −23.8 | −24.9 |
| Dy | −0.6 | −21.6 | −22.2 | −26.3 |
| Ho | 4.1 | −24.5 | −20.4 | −27.6 |
| Er | 8.7 | −27.5 | −18.8 | −28.7 |
| Tm | 14.8 | −32.1 | −17.4 | −28.6 |
| Yb | 20.0 | −35.2 | −15.2 | −29.8 |
| Lu | 25.2 | −38.7 | −13.5 | −31.1 |

**Table A7.** Differences of free reaction enthalpies $\Delta\Delta_r G$ of three $REHO$ synthesis approaches.

| | $\Delta\Delta_r G/\text{kJ mol}^{-1}$ | | |
|---|---|---|---|
| *RE* | $RE_2O_3 + 3\,CaH_2$ | $2\,REH_3 + 3\,CaO$ | $3\,REOF + 3\,LiH$ |
| Y | −40.3 | −54.5 | −74.7 |
| La | −31.0 | −32.1 | −47.6 |
| Ce | 12.8 | −41.1 | −34.7 |
| Pr | 4.5 | −34.6 | −32.4 |
| Nd | −2.7 | −29.4 | −30.7 |
| Pm | −10.6 | −24.0 | −29.3 |
| Sm | −15.4 | −20.5 | −28.1 |
| Eu | −23.2 | −15.3 | −26.9 |
| Gd | −31.2 | −9.8 | −25.4 |
| Tb | −37.3 | −5.2 | −23.8 |
| Dy | −43.1 | −0.6 | −22.2 |
| Ho | −49.0 | 4.1 | −20.4 |
| Er | −55.0 | 8.7 | −18.8 |
| Tm | −64.2 | 14.8 | −17.4 |
| Yb | −70.5 | 20.0 | −15.2 |
| Lu | −77.3 | 25.2 | −13.5 |

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
