# Peer review of "Computational Chemistry-Guided Syntheses and Crystal Structures of the Heavier Lanthanide Hydride Oxides DyHO, ErHO, and LuHO"

_crystals, doi:10.3390/cryst11070750_

Round 1
Reviewer 1 Report
It is very solid joint experimental and computational work
on synthesis and structural characterizations of the new heavier lanthanide hydride oxides. It is well written and the valuable outcomes are interesting for the crystallographic community; from this respect this work shall be published upon minor revisions noted below.
1) Please comment on (prospective) properties of the reported species. Explain
to readers why such species are important.
2) Please comment on why REHO are more stable for the larger rare earth elements? any role of entropic factor?
3) Please comment on novelties with respect to Naturforsch. B 2018, 73, 535.
4) Please justify the applied computational details.
Author Response
Reviewer 1:
It is very solid joint experimental and computational work
on synthesis and structural characterizations of the new heavier lanthanide hydride oxides. It is well written and the valuable outcomes are interesting for the crystallographic community; from this respect this work shall be published upon minor revisions noted below.
1) Please comment on (prospective) properties of the reported species. Explain
to readers why such species are important.
Thanks for the hint. We added two sentences in the conclusion that address the compounds properties and the importance of our study in context of ion conductors and catalysts in systems RE-H-O.
2) Please comment on why REHO are more stable for the larger rare earth elements? any role of entropic factor?
Thanks for the hint. The increased stability of the larger lanthanides probably results from softer cations, a hypothesis we added in a sentence on p. 17. The entropy does not play a role in the DFT calculations, as all models assume a temperature of zero kelvin.
3) Please comment on novelties with respect to Naturforsch. B 2018, 73, 535.
Thank you for your comment. We highlighted the motivation of this paper as a successor to our last articles in the second to last paragraph of the introduction on p. 4. In addition to Naturforsch B 2018, 73, 535, we included neutron diffraction studies on DyHO, ErHO and two LuHO polymorphs. The DFT section includes different reaction pathways and also concentrates on possible side reactions.
4) Please justify the applied computational details.
Thank you for your comment. The employed computational methods are in accordance with current literature reports on related topics (Liu et al., RSC Adv. 2016, 6, 9822; Yamashita et al., J. Am. Chem. Soc. 2017, 139, 18240) and are therefore easy transferable. All parameters were well converged.
Reviewer 2 Report
The manuscript by Zapp et al. is an interesting contribution to the field of mixed anion materials and hydride oxides/oxyhydrides. The manuscript brings new knowledge on the LnHO series and I can recommend publication after some adjustments:
It is not communicated clearly that there are two polymorphs of LuHO, and it must be communicated more clearly when a phase is an anion ordered or not. It is not always easy to understand if the phase that is discussed is ordered or not without going back in the manuscript to check. By adjusting this, the manuscript will be much easier to follow for readers. In this context, Figure 1 does not do a very good job to display the differences in the crystal structures. This figure should be revised, and missing axis crosses should be added. I would encourage to consider describing the ordering schemes in the coordination polyhedra of Ln.
Section 3.3 has a problem; it presents the anion ordering in LnHO compounds as mostly unknown. LaHO and NdHO are known to be ordered tetragonal and are therefore difficult to add to the trend, but to use the cubic reports is not better. This is because deviation from tetragonal symmetry indicates off-stoichiometry. The reason for the tetragonal distortion is explained well by Yamashita et al.(https://doi.org/10.1021/jacs.8b06187). Smaller Ln than Sm, on the other hand, are reported to be cubic. The reports are not coherent with respect to ordering, but this is can be attributed to the fact that X-rays are used in many of the studies, which does not differentiate between the ordered (F-43m) and disordered (Fm-3m). X-ray data cannot be used for this discussion as you don’t know the ordering and therefore cannot trust the stoichiometry. However, it is interesting to discuss how off-stoichiometry influences anion ordering. But it needs to be done within a different framework.
Are there any differences between the cubic and orthorhombic LuHO in stoichiometry? Why is one orthorhombic? Can this be justified based on the arguments in https://doi.org/10.1021/jacs.8b06187 ?
A shortcoming of the 3.4 is that “X Liu, TS Bjørheim, R Haugsrud RSC advances 6 (12), 9822-9826 2016” (https://doi.org/10.1039/C5RA26552E) is not discussed. In this paper, other formation reactions of the LnHO are discussed, and this should be discussed in the manuscript.
Raman spectroscopy is successfully utilized for NdHO (https://doi.org/10.1016/j.jssc.2011.05.025). It is therefore a suitable technique to identify the symmetry of LnHO.
Table 3, it is not initially obvious what parameters are from XRD and from neutron diffraction. Please make the tables with both XRD and neutron diffraction so they cannot be misinterpreted.
For Table 4, it is unclear why the cubic LuHO phase has several different bond lengths for Lu-O/H, in the cubic phase, these distances should be identical. The cif file in supplemental also suggests this.
For the conclusion, you have four hydride oxides, not five.
Author Response
Reviewer 2:
The manuscript by Zapp et al. is an interesting contribution to the field of mixed anion materials and hydride oxides/oxyhydrides. The manuscript brings new knowledge on the LnHO series and I can recommend publication after some adjustments:
It is not communicated clearly that there are two polymorphs of LuHO, and it must be communicated more clearly when a phase is an anion ordered or not. It is not always easy to understand if the phase that is discussed is ordered or not without going back in the manuscript to check. By adjusting this, the manuscript will be much easier to follow for readers. In this context, Figure 1 does not do a very good job to display the differences in the crystal structures. This figure should be revised, and missing axis crosses should be added. I would encourage to consider describing the ordering schemes in the coordination polyhedra of Ln.
Thank you for your comment. We improved the paragraphs on both polymorphs of LuHO to make the reader understand their differences easier. We updated Figure 1 and added a sentence about the coordination polyhedra in LuDO.
Section 3.3 has a problem; it presents the anion ordering in LnHO compounds as mostly unknown. LaHO and NdHO are known to be ordered tetragonal and are therefore difficult to add to the trend, but to use the cubic reports is not better. This is because deviation from tetragonal symmetry indicates off-stoichiometry. The reason for the tetragonal distortion is explained well by Yamashita et al.(https://doi.org/10.1021/jacs.8b06187). Smaller Ln than Sm, on the other hand, are reported to be cubic. The reports are not coherent with respect to ordering, but this is can be attributed to the fact that X-rays are used in many of the studies, which does not differentiate between the ordered (F-43m) and disordered (Fm-3m). X-ray data cannot be used for this discussion as you don’t know the ordering and therefore cannot trust the stoichiometry. However, it is interesting to discuss how off-stoichiometry influences anion ordering. But it needs to be done within a different framework.
Thanks for your comment. The section 3.3 shows for the first time, that it is difficult to compare cubic REHO compounds, because some of them are REH1+2xO1-x instead of REHO. We show, that the lattice parameter might be used to differentiate both compound classes, which is challenging with other methods. We agree that a systematic study on the phase width in the system RE-H-O with respective neutron diffraction analysis and correlation with the cationic radii of RE is still missing, this is however beyond the scope of our work. For a better understanding, we restructured the discussion of Figure 7 and Table 6 on p. 13 and changed the misleading term “unknown” in figure 7 to “assumed disordered” and “assumed ordered”.
Are there any differences between the cubic and orthorhombic LuHO in stoichiometry? Why is one orthorhombic? Can this be justified based on the arguments in https://doi.org/10.1021/jacs.8b06187 ?
Thank you for the comment. Both polymorphs do not deviate in stoichiometry. The arguments shown in Yamashita et al.’s paper cannot be applied to the anti-LiMgN structure type. It is known for RE = Y (medium sized cation) and Lu (small cation), but not for Dy – Tm (medium – small cations). We currently assume, that the preferred polymorph is correlated with the polarizability of the cations; however, there is not sufficient evidence for reliable assumptions yet. We added a discussion of this on p. 10.
A shortcoming of the 3.4 is that “X Liu, TS Bjørheim, R Haugsrud RSC advances 6 (12), 9822-9826 2016” (https://doi.org/10.1039/C5RA26552E) is not discussed. In this paper, other formation reactions of the LnHO are discussed, and this should be discussed in the manuscript.
Thanks for the hint. We added a reference to this study on p. 17.
Raman spectroscopy is successfully utilized for NdHO (https://doi.org/10.1016/j.jssc.2011.05.025). It is therefore a suitable technique to identify the symmetry of LnHO.
This is correct, we added the reference on p. 13. Our experiments with an excitation wavelength of 532 nm did however not produce conclusive results.
Table 3, it is not initially obvious what parameters are from XRD and from neutron diffraction. Please make the tables with both XRD and neutron diffraction so they cannot be misinterpreted.
Thanks for the hint. We split table 3 into two tables (now table 3 and 4).
For Table 4, it is unclear why the cubic LuHO phase has several different bond lengths for Lu-O/H, in the cubic phase, these distances should be identical. The cif file in supplemental also suggests this.
Thank you for the comment. The atomic distances of cubic LuHO are provided in table A2, table 4 (now table 5) shows the atomic distances of the orthorhombic polymorphs.
For the conclusion, you have four hydride oxides, not five.
Thank you for the comment. We counted orthorhombic LuHO and LuDO as two hydride oxides, therefore it’s five in total (DyHO, ErHO, orthorhombic LuHO, orthorhombic LuDO, cubic LuHO).
Round 2
Reviewer 2 Report
The authors have adjusted the manuscript according to my comments. I found a mistake in Table 4, where the hydride site is not given. When this is corrected, the manuscript can be accepted for publication.
Author Response
Tab. 4 reports on refinements based on X-ray data, where the contribution of H atoms is negligible. It has therefore not been included in the model. A typo was corrected in the legend to Tab. 4 ("on" added).